# Deconvolving Feedback Loops
# in Recommender Systems

**Ayan Sinha**
Purdue University
sinhayan@mit.edu

**David F. Gleich**
Purdue University
dgleich@purdue.edu

**Karthik Ramani**
Purdue University
ramani@purdue.edu

## Abstract

Collaborative filtering is a popular technique to infer users' preferences on new content based on the collective information of all users preferences. Recommender systems then use this information to make personalized suggestions to users. When users accept these recommendations it creates a feedback loop in the recommender system, and these loops iteratively influence the collaborative filtering algorithm's predictions over time. We investigate whether it is possible to identify items affected by these feedback loops. We state sufficient assumptions to deconvolve the feedback loops while keeping the inverse solution tractable. We furthermore develop a metric to unravel the recommender system's influence on the entire user-item rating matrix. We use this metric on synthetic and real-world datasets to (1) identify the extent to which the recommender system affects the final rating matrix, (2) rank frequently recommended items, and (3) distinguish whether a user's rated item was recommended or an intrinsic preference. Our results indicate that it is possible to recover the ratings matrix of intrinsic user preferences using a single snapshot of the ratings matrix without any temporal information.

## 1 Introduction

Recommender systems have been helpful to users for making decisions in diverse domains such as movies, wines, food, news among others [19, 23]. However, it is well known that the interface of these systems affect the users' opinion, and hence, their ratings of items [7, 24].Thus, broadly speaking, a user's rating of an item is either his or her intrinsic preference or the influence of the recommender system (RS) on the user [2]. As these ratings implicitly affect recommendations to other users through feedback, it is critical to quantify the role of feedback in content personalization [22]. *Thus the primary motivating question for this paper is: Given only a user-item rating matrix, is it possible to infer whether any preference values are influenced by a RS? Secondary questions include: Which preference values are influenced and to what extent by the RS? Furthermore, how do we recover the true preference value of an item to a user?*

We develop an algorithm to answer these questions using the singular value decomposition (SVD) of the observed ratings matrix (Section 2). The genesis of this algorithm follows by viewing the observed ratings at any point of time as union of true ratings and recommendations:

$$R_{\text{obs}} = R_{\text{true}} + R_{\text{recom}} \tag{1}$$

where $R_{\text{obs}}$ is the observed rating matrix at a given instant of time, $R_{\text{true}}$ is the rating matrix due to users' true preferences of items (along with any external influences such as ads, friends, and so on) and $R_{\text{recom}}$ is the rating matrix which indicates the RS's contribution to the observed ratings. Our more formal goal is to recover $R_{\text{true}}$ from $R_{\text{obs}}$. But this is impossible without strong modeling assumptions; any rating is just as likely to be a true rating as due to the system.

Thus, we make strong, but plausible assumptions about a RS. In essence, these assumptions prescribe a precise model of the recommender and prevent its effects from completely dominating the future.

With these assumptions, we are able to mathematically relate $\boldsymbol{R}_{\text{true}}$ and $\boldsymbol{R}_{\text{obs}}$. This enables us to find the centered rating matrix $\boldsymbol{R}_{\text{true}}$ (up to scaling). We caution readers that these assumptions are designed to create a model that we can tractably analyze, and they should not be considered *limitations* of our ideas. Indeed, the strength of this simplistic model is that we can use its insights and predictions to analyze far more complex real-world data. One example of this model is that the notion of $\boldsymbol{R}_{\text{true}}$ is a convenient fiction that represents some idealized, unperturbed version of the ratings matrix. Our model and theory suggests that $\boldsymbol{R}_{\text{true}}$ ought to have some relationship with the observed ratings, $\boldsymbol{R}_{\text{obs}}$. By studying these relationships, we will show that we gain useful insights into the strength of various feedback and recommendation processes in real-data.

In that light, we use our theory to develop a heuristic, but accurate, metric to quantitatively infer the influence of a RS (or any set of feedback effects) on a ratings matrix (Section 3). Additionally, we propose a metric for evaluating the influence of a recommender system on each user-item rating pair. Aggregating these scores over all users helps identify putative highly recommended items. The final metrics for a RS provide insight into the quality of recommendations and argue that Netflix had a better recommender than MovieLens, for example. This score is also sensitive to all cases where we have ground-truth knowledge about feedback processes akin to recommenders in the data.

## 2 Deconvolving feedback

We first state equations ans assumptions under which the true rating matrix is recoverable (or deconvolvable) from the observed matrix, and provide an algorithm to deconvolve using the SVD.

### 2.1 A model recommender system

Consider a ratings matrix $\boldsymbol{R}$ of dimension $m \times n$ where $m$ is the number of users and $n$ is the number of items being rated. Users are denoted by subscript $u$, and items are denoted by subscript $i$, i.e., $\boldsymbol{R}_{u,i}$ denotes user $u$'s rating for item $i$. As stated after equation (1), our objective is to decouple $\boldsymbol{R}_{\text{true}}$ from $\boldsymbol{R}_{\text{recom}}$ given the matrix $\boldsymbol{R}_{\text{obs}}$. Although this problem seems intractable, we list a series of assumptions under which a closed form solution of $\boldsymbol{R}_{\text{true}}$ is deconvolvable from $\boldsymbol{R}_{\text{obs}}$ alone.

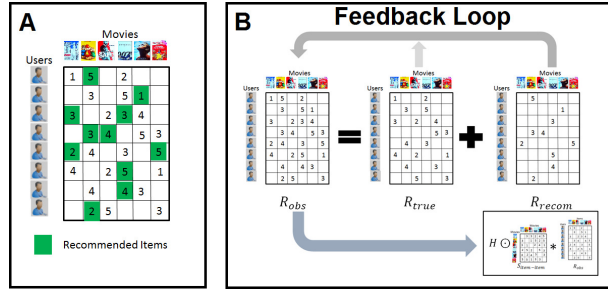

Figure 1: Subfigure A shows a ratings matrix with recommender induced ratings and true ratings; Figure B: Feedback loop in RS wherein the observed ratings is a function of the true ratings and ratings induced by a RS

**Assumption 1** *The feedback in the RS occurs through the iterative process involving the observed ratings and an item-item similarity matrix $\boldsymbol{S}$:* [1]

$$\boldsymbol{R}_{obs} = \boldsymbol{R}_{true} + \boldsymbol{H} \odot (\boldsymbol{R}_{obs}\boldsymbol{S}). \tag{2}$$

Here $\odot$ indicates Hadamard, or entrywise product, given as: $(\boldsymbol{H} \odot \boldsymbol{R})_{u,i} = \boldsymbol{H}_{u,i} \cdot \boldsymbol{R}_{u,i}$. This assumption is justified because in many collaborative filtering techniques, $\boldsymbol{R}_{\text{recom}}$ is a function of the observed ratings $\boldsymbol{R}_{\text{obs}}$ and the item-item similarity matrix, $\boldsymbol{S}$. The matrix $\boldsymbol{H}$ is an indicator matrix over a set of items where the user followed the recommendation and agreed with it. This matrix is essentially completely unknown and is essentially unknowable without direct human interviews. The model RS equation (2) then iteratively updates $\boldsymbol{R}_{\text{obs}}$ based on commonly rated items by users. This key idea is illustrated in Figure 1. The recursion progressively fills all missing entries in matrix $\boldsymbol{R}_{\text{obs}}$ starting from $\boldsymbol{R}_{\text{true}}$. The recursions do not update $\boldsymbol{R}_{\text{true}}$ in our model of a RS. If we were to explicitly consider the state of matrix $\boldsymbol{R}_{\text{obs}}$ after $k$ iterations, $\boldsymbol{R}_{\text{obs}}^{k+1}$ we get:

$$\boldsymbol{R}_{\text{obs}}^{k+1} = \boldsymbol{R}_{\text{true}} + \boldsymbol{H}^{(k)} \odot (\boldsymbol{R}_{\text{obs}}^{k}\boldsymbol{S}_k) = \boldsymbol{R}_{\text{true}} + \boldsymbol{H}^{(k)} \odot \left((\boldsymbol{R}_{\text{true}} + \boldsymbol{H}^{(k-1)} \odot (\boldsymbol{R}_{\text{obs}}^{k-1}\boldsymbol{S}_{k-1}))\boldsymbol{S}_k\right) = \dots \tag{3}$$

Here $\boldsymbol{S}_k$ is the item-item similarity matrix induced by the observed matrix at state $k$. The above equation 3 is naturally initialized as $\boldsymbol{R}_{\text{obs}}^{1} = \boldsymbol{R}_{\text{true}}$ along with the constraint $\boldsymbol{S}_1 = \boldsymbol{S}_{\text{true}}$, i.e, the similarity

matrix at the first iteration is the similarity matrix induced by the matrix of true preferences, $\mathbf{R}_{\text{true}}$. Thus, we see that $\mathbf{R}_{\text{obs}}$ is an implicit function of $\mathbf{R}_{\text{true}}$ and the set of similarity matrices $\mathbf{S}_k, \mathbf{S}_{k-1}, \ldots \mathbf{S}_1$.

**Assumption 2** *Hadamard product $\mathbf{H}^{(k)}$ is approximated with a probability parameter $\alpha_k \in (0, 1]$.*

We model the selection matrix $\mathbf{H}^{(k)}$ and it's Hadamard problem in expectation and replace the successive matrices $\mathbf{H}^{(k)}$ with independent Bernoulli random matrices with probability $\alpha_k$. Taking the expectation allows us to replace the matrix $\mathbf{H}^{(k)}$ with the probability parameter $\alpha_k$ itself:

$$\mathbf{R}_{\text{obs}}^{k+1} = \mathbf{R}_{\text{true}} + \alpha_k(\mathbf{R}_{\text{obs}}^k \mathbf{S}_k) = \mathbf{R}_{\text{true}} + \alpha_k\big((\mathbf{R}_{\text{true}} + \alpha_{k-1}(\mathbf{R}_{\text{obs}}^{k-1}\mathbf{S}_{k-1}))\mathbf{S}_k\big) = \ldots \qquad (4)$$

The set of $\mathbf{S}_k, \mathbf{S}_{k-1}, \cdots$ are *apriori* unknown. We are now faced with the task of constructing a valid similarity metric. Towards this end, we make our next assumption.

**Assumption 3** *The user mean $\bar{\mathbf{R}}_u$ in the observed and true matrix are roughly equal: $\bar{\mathbf{R}}_u^{(obs)} \approx \bar{\mathbf{R}}_u^{(true)}$. The Euclidean item norms $\|\mathbf{R}_i\|$ are also roughly equal: $\|\mathbf{R}_i^{(obs)}\| \approx \|\mathbf{R}_i^{(true)}\|$.*

These assumptions are justified because ultimately we are interested in relative preferences of items for a user and unbiased relative ratings of items by users. These can be achieved by centering users and the normalizing item ratings, respectively, in the true and observed ratings matrices. We quantitatively investigate this assumption in the supplementary material. Using this assumption, the similarity metric then becomes:

$$S(i, j) = \frac{\sum_{u \in U}(\mathbf{R}_{u,i} - \bar{\mathbf{R}}_u)(\mathbf{R}_{u,j} - \bar{\mathbf{R}}_u)}{\sqrt{\sum_{u \in U}(\mathbf{R}_{u,i} - \bar{\mathbf{R}}_u)^2}\sqrt{\sum_{u \in U}(\mathbf{R}_{u,j} - \bar{\mathbf{R}}_u)^2}} \qquad (5)$$

This metric is known as the adjusted cosine similarity, and preferred over cosine similarity because it mitigates the effect of rating schemes over users [25]. Using the relations $\tilde{\mathbf{R}}_{u,i} = \mathbf{R}_{u,i} - \bar{\mathbf{R}}_u$, and, $\hat{\mathbf{R}}_{u,i} = \frac{\tilde{\mathbf{R}}_{u,i}}{\|\tilde{\mathbf{R}}_i\|} = \frac{\mathbf{R}_{u,i} - \bar{\mathbf{R}}_u}{\sqrt{\sum_{u \in U}(\mathbf{R}_{u,i} - \bar{\mathbf{R}}_u)^2}}$, the expression of our recommender (4) becomes:

$$\hat{\mathbf{R}}_{\text{obs}} = \hat{\mathbf{R}}_{\text{true}}(\mathbf{I} + f_1(\mathbf{a}_1)\hat{\mathbf{R}}_{\text{true}}^T\hat{\mathbf{R}}_{\text{true}} + f_2(\mathbf{a}_2)(\hat{\mathbf{R}}_{\text{true}}^T\hat{\mathbf{R}}_{\text{true}})^2 + f_3(\mathbf{a}_3)(\hat{\mathbf{R}}_{\text{true}}^T\hat{\mathbf{R}}_{\text{true}})^3 + \ldots) \qquad (6)$$

Here, $f_1, f_2, f_3 \ldots$ are functions of the probability parameters $\mathbf{a}_k = [\alpha_1, \alpha_2, \ldots \alpha_k, \ldots]$ of the form $f_z(\mathbf{a}_z) = c\alpha_1^{c_1}\alpha_1^{c_2} \ldots \alpha_k^{c_k} \ldots$ such that $\sum_k c_k = z$, and $c$ is a constant. The proof of equation 6 is in the supplementary material. We see that the centering and normalization results in $\hat{\mathbf{R}}_{\text{obs}}$ being explicitly represented in terms of $\hat{\mathbf{R}}_{\text{true}}$ and coefficients $f(\mathbf{a})$. It is now possible to recover $\hat{\mathbf{R}}_{\text{true}}$, but the coefficients $f(\mathbf{a})$ are *apriori* unknown. Thus, our next assumption.

**Assumption 4** $f_z(\mathbf{a}_z) = \alpha^z$, *i.e., the coefficients of the series (6) are induced by powers of a constant probability parameter $\alpha \in (0, 1]$.*

Note that in recommender (3), $\mathbf{R}_{\text{obs}}$ becomes denser with every iteration, and hence the higher order Hadamard products in the series fill fewer missing terms. The effect of absorbing the unknowable probability parameters, $\alpha_k$'s into single probability parameter $\alpha$ is similar. Powers of $\alpha$, produce successively less of an impact, just as in the true model. The governing expression now becomes:

$$\hat{\mathbf{R}}_{\text{obs}} = \hat{\mathbf{R}}_{\text{true}}(\mathbf{I} + \alpha\hat{\mathbf{R}}_{\text{true}}^T\hat{\mathbf{R}}_{\text{true}} + \alpha^2(\hat{\mathbf{R}}_{\text{true}}^T\hat{\mathbf{R}}_{\text{true}})^2 + \alpha^3(\hat{\mathbf{R}}_{\text{true}}^T\hat{\mathbf{R}}_{\text{true}})^3 + \ldots) \qquad (7)$$

In order to ensure convergence of this equation, we make our final assumption.

**Assumption 5** *The spectral radius of the similarity matrix $\alpha\hat{\mathbf{R}}_{true}^T\hat{\mathbf{R}}_{true}$ is less than 1.*

This assumption enables us to write the infinite series representing $\hat{\mathbf{R}}_{\text{obs}}$, $\hat{\mathbf{R}}_{\text{true}}(\mathbf{I} + \alpha\hat{\mathbf{R}}_{\text{true}}^T\hat{\mathbf{R}}_{\text{true}} + \alpha^2(\hat{\mathbf{R}}_{\text{true}}^T\hat{\mathbf{R}}_{\text{true}})^2 + \alpha^3(\hat{\mathbf{R}}_{\text{true}}^T\hat{\mathbf{R}}_{\text{true}})^3 + \ldots)$ as $(1 - \alpha\hat{\mathbf{R}}_{\text{true}}^T\hat{\mathbf{R}}_{\text{true}})^{-1}$. It states that given $\alpha$, we scale the matrix $\hat{\mathbf{R}}_{\text{true}}^T\hat{\mathbf{R}}_{\text{true}}$ such that the spectral radius of $\alpha\hat{\mathbf{R}}_{\text{true}}^T\hat{\mathbf{R}}_{\text{true}}$ is less than 1 [2]. Then we are then able to recover $\hat{\mathbf{R}}_{\text{true}}^T$ up to a scaling constant.

**Discussion of assumptions.** We now briefly discuss the implications of our assumptions. First, assumption 1 states the recommender model. Assumption 2 states that we are modeling *expected*

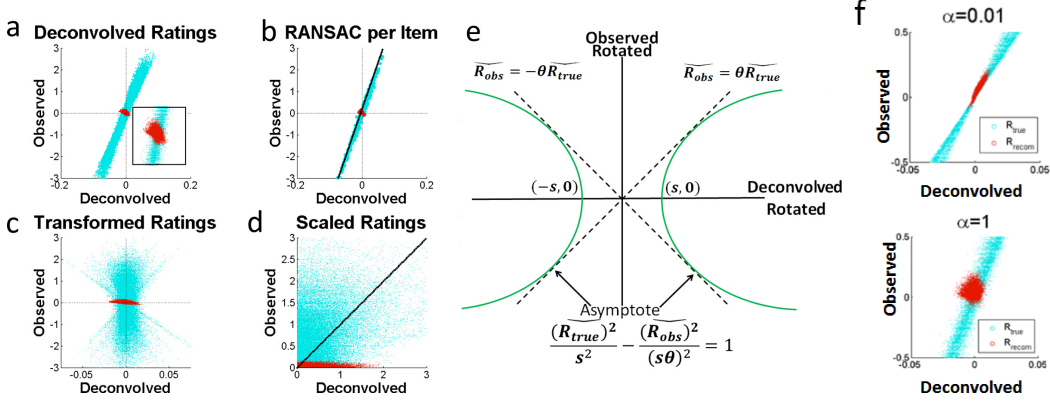

Figure 2: (a) to (f): Our procedure for scoring ratings based on the deconvolved scores with true initial ratings in cyan and ratings due to recommender in red. (a) The observed and deconvolved ratings. (b) The RANSAC fit to extract straight line passing through data points for each item. (c) Rotation and translation of data points using fitted line such that the scatter plot is approximately parallel to $y$-axis and recommender effects are distinguishable along x-axis. (d) Scaling of data points used for subsequent score assignment. (e) Score assignment using the vertex of the hyperbola with slope $\theta = 1$ that passes through the data point. (f) Increasing $\alpha$ deconvolves implicit feedback loops to a greater extent and better discriminates recommender effects as illustrated by the red points which show more pronounced deviation when $\alpha = 1$.

*behavior* rather than *actual behavior*. Assumptions 3-5 are key to our method working. They essentially state that the RS's effects are limited in scope so that they cannot dominate the world. This has a few interpretations on real-world data. The first would be that we are considering the impact of the RS over a short time span. The second would be that the recommender effects are essentially *second-order* and that there is some other *true effect* which dominates them. We discuss the mechanism of solving equation 7 using the above set of five assumptions next.

## 2.2 The algorithm for deconvolving feedback loops

**Theorem 1** *Assuming the RS follows* (7), $\alpha$ *is between 0 and* 1, *and the singular value decomposition of the observed rating matrix is,* $\hat{\boldsymbol{R}}_{obs} = \boldsymbol{U}\boldsymbol{\Sigma}_{obs}\boldsymbol{V}^T$, *the deconvolved matrix* $\boldsymbol{R}_{true}$ *of true ratings is given as* $\boldsymbol{U}\boldsymbol{\Sigma}_{true}\boldsymbol{V}^T$, *where the* $\boldsymbol{\Sigma}_{true}$ *is a diagonal matrix with elements:*

$$\sigma_i^{true} = \frac{-1}{2\alpha\sigma_i^{obs}} + \sqrt{\frac{1}{4\alpha^2(\sigma_i^{obs})^2} + \frac{1}{\alpha}} \tag{8}$$

The proof of the theorem is in the supplementary material. In practical applications, the feedback loops are deconvolved by taking a truncated-SVD (low rank approximation) instead of the complete decomposition. In this process, we naturally concede accuracy for performance. We consider the matrix of singular values $\tilde{\boldsymbol{\Sigma}}_{obs}$ to only contain the $k$ largest singular values (the other singular values are replaced by zero). We now state Algorithm 1 for deconvolving feedback loops. The algorithm is simple to compute as it just involves a singular value decomposition of the observed ratings matrix.

## 3 Results and recommender system scoring

We tested our approach for deconvolving feedback loops on synthetic RS, and designed a metric to identify the ratings most affected by the RS. We then use the same automated technique to study real-world ratings data, and find that the metric is able to identify items influenced by a RS.

---
**Algorithm 1** Deconvolving Feedback Loops
---
**Input:** $\boldsymbol{R}_{\text{obs}}, \alpha, k$, where $\boldsymbol{R}_{\text{obs}}$ is observed ratings matrix, $\alpha$ is parameter governing feedback loops and $k$ is number of singular values

**Output:** $\hat{\boldsymbol{R}}_{\text{true}}$, True rating matrix

1: Compute $\tilde{\boldsymbol{R}}_{\text{obs}}$ given $\boldsymbol{R}_{\text{obs}}$, where $\tilde{\boldsymbol{R}}_{\text{obs}}$ is user centered observed matrix
2: Compute $\hat{\boldsymbol{R}}_{\text{obs}} \leftarrow \tilde{\boldsymbol{R}}_{\text{obs}} D_N^{-1}$, where $\hat{\boldsymbol{R}}_{\text{obs}}$ is item-normalized rating matrix, and $D_N^{-1}$ is diagonal matrix of item-norms
$$D_N(i,i) = \sqrt{\sum_{u \in U}(\boldsymbol{R}_{u,i} - \bar{\boldsymbol{R}}_u)^2}$$
3: Solve $U\Sigma_{\text{obs}}V^T \leftarrow SVD(\hat{\boldsymbol{R}}_{\text{obs}}, k)$, the truncated SVD corresponding to $k$ largest singular values.
4: Perform $\sigma_i^{\text{true}} \leftarrow (\frac{-1}{2\alpha\sigma_i^{\text{obs}}} + \sqrt{\frac{1}{4\alpha^2(\sigma_i^{\text{obs}})^2} + \frac{1}{\alpha}})$ for all $i$
5: **return** $U, \Sigma_{\text{true}}, V^T$
---

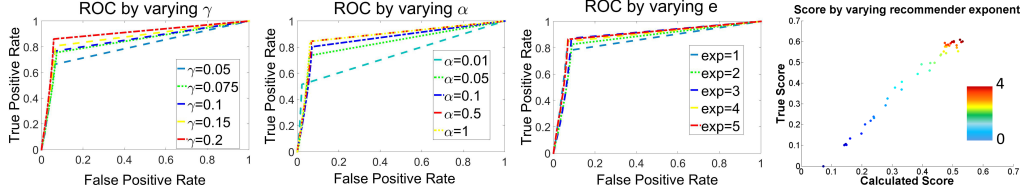

Figure 3: Results for a synthetic RS with controllable effects. (Left to right): (a) ROC curves by varying data sparsity (b) ROC curves by varying the parameter $\alpha$ (c) ROC curves by varying feedback exponent (d) Score assessing the overall recommendation effects as we vary the true effect.

## 3.1 Synthetic data simulating a real-world recommender system

We use item response theory to generate a sparse true rating matrix $\boldsymbol{R}_{\text{true}}$ using a model related to that in [12]. Let $a_u$ be the center of user $u$'s rating scale, and $b_u$ be the rating sensitivity of user $u$. Let $t_i$ be the intrinsic score of item $i$. We generate a user-item rating matrix as:

$$\boldsymbol{R}_{u,i} = L[a_u + b_u t_i + \eta_{u,i}] \tag{9}$$

where $L[\omega]$ is the discrete levels function assigning a score in the range 1 to 5: $L[\omega] = \max(\min(\text{round}(\omega), 5), 1)$ and $\eta_{u,i}$ is a noise parameter. In our experiment, we draw $a_u \sim N(3, 1)$, $b_u \sim N(0.5, 0.5)$, $t_u \sim N(0.1, 1)$, and $\eta_{u,i} \sim \epsilon N(0, 1)$, where $N$ is a standard normal, and $\epsilon$ is a noise parameter. We sample these ratings uniformly at random by specifying a desired level of rating sparsity $\gamma$ which serves as the input, $\boldsymbol{R}_{\text{true}}$, to our RS. We then run a cosine similarity based RS, progressively increasing the density of the rating matrix. The unknown ratings are iteratively updated using the standard item-item collaborative filtering technique [8] as $\boldsymbol{R}_{u,i}^{k+1} = \frac{\sum_{j \in i}(s_{i,j}^k \boldsymbol{R}_{u,j}^k)}{\sum_{j \in i}(|s_{i,j}^k|)}$, where $k$ is the iteration number and $\boldsymbol{R}^0 = \boldsymbol{R}_{\text{true}}$, and the similarity measure at the $k^{th}$ iteration is given as $s_{i,j}^k = \frac{\sum_{u \in U} \boldsymbol{R}_{u,i}^k \boldsymbol{R}_{u,j}^k}{\sqrt{\sum_{u \in U}(\boldsymbol{R}_{u,i}^k)^2}\sqrt{\sum_{u \in U}(\boldsymbol{R}_{u,j}^k)^2}}$. After the $k^{th}$ iteration, each synthetic user accepts the top $r$ recommendations with probability proportional to $(\boldsymbol{R}_{u,i}^{k+1})^e$, where $e$ is an exponent controlling the frequency of acceptance. We fix the number of iterative updates to be 10, $r$ to be 10 and the resulting rating matrix is $\boldsymbol{R}_{\text{obs}}$. We deconvolve $\boldsymbol{R}_{\text{obs}}$ as per Algorithm 1 to output $\hat{\boldsymbol{R}}_{\text{true}}$. Recall, $\hat{\boldsymbol{R}}_{\text{true}}$ is user-centered and item-normalized. In the absence of any recommender effects $\boldsymbol{R}_{\text{recom}}$, the expectation is that $\hat{\boldsymbol{R}}_{\text{true}}$ is perfectly correlated with $\hat{\boldsymbol{R}}_{\text{obs}}$. The absence of a linear correlation hints at factors extraneous to the user, i.e., the recommender. Thus, we plot $\hat{\boldsymbol{R}}_{\text{true}}$ (the deconvolved ratings) against the $\hat{\boldsymbol{R}}_{\text{obs}}$, and search for characteristic signals that exemplify recommender effects (see Figure 2a and inset).

## 3.2 A metric to assess a recommender system

We develop an algorithm guided by the intuition that deviation of ratings from a straight line suggest recommender effects (Algorithm 2). The procedure is visually elucidated in Figure 2. We consider fitting a line to the observed and deconvolved (equivalently estimated true) ratings; however, our experiments indicate that least square fit of a straight line in the presence of severe recommender effects is not robust. The outliers in our formulation correspond to recommended items. Hence, we use random sample consensus or the RANSAC method [11] to fit a straight line *on a per item basis*

Table 1: Datasets and parameters

| Dataset | Users | Items | Min RPI | Rating | k in SVD | Score |
|---|---|---|---|---|---|---|
| Jester-1 | 24.9K | 100 | 1 | 615K | 100 | 0.0487 |
| Jester-2 | 50.6K | 140 | 1 | 1.72M | 140 | 0.0389 |
| MusicLab-Weak | 7149 | 48 | 1 | 25064 | 48 | 0.1073 |
| MusicLab-Strong | 7192 | 48 | 1 | 23386 | 48 | 0.1509 |
| MovieLens-100K | 943 | 603 | 50 | 83.2K | 603 | 0.2834 |
| MovieLens-1M | 6.04K | 2514 | 50 | 975K | 2514 | 0.3033 |
| MovieLens-10M | 69.8K | 7259 | 50 | 9.90M | 1500 | 0.3821 |
| BeerAdvocate | 31.8K | 9146 | 20 | 1.35M | 1500 | 0.2223 |
| RateBeer | 28.0K | 20129 | 20 | 2.40M | 1500 | 0.1526 |
| Fine Foods | 130K | 5015 | 20 | 329K | 1500 | 0.1209 |
| Wine Ratings | 21.0K | 8772 | 20 | 320K | 1500 | 0.1601 |
| Netflix | 480K | 16795 | 100 | 100M | 1500 | 0.2661 |

(Figure 2b). All these straight lines are translated and rotated so as to coincide with the y-axis as displayed in Figure 2c. Observe that the data points corresponding to recommended ratings pop out as a bump along the x-axis. Thus, the effect of the RANSAC and rotation is to place the ratings into a precise location. Next, the ratings are scaled so as to make the maximum absolute values of the rotated and translated $\breve{R}_{\text{true}}, \breve{R}_{\text{obs}}$, values to be equal (Figure 2d).

The scores we design are to measure "extent" into the $x$-axis. But we want to consider some allowable vertical displacement. The final score we assign is given by fitting a hyperbola through each rating viewed as a point: $\breve{R}_{\text{true}}, \breve{R}_{\text{obs}}$. A straight line of slope, $\theta = 1$ passing through the origin is fixed as an asymptote to all hyperbolas. The vertex of this hyperbola serves as the score of the corresponding data point. The higher the value of the vertex of the associated hyperbola to a data point, the more likely is the data point to be recommended item. Using the relationship between slope of asymptote, and vertex of hyperbola, the score $s(\breve{R}_{\text{true}}, \breve{R}_{\text{obs}})$ is given by:

$$s(\breve{R}_{\text{true}}, \breve{R}_{\text{obs}}) = \text{real}(\sqrt{\breve{R}_{\text{true}}^2 - \breve{R}_{\text{obs}}^2}) \tag{10}$$

We set the slope of the asymptote, $\theta = 1$, because the maximum magnitudes of $\breve{R}_{\text{true}}, \breve{R}_{\text{obs}}$ are equal (see Figure 2 d,e). The overall algorithm is stated in the supplementary material. Scores are zero if the point is inside the hyperbola with vertex 0.

## 3.3 Identifying high recommender effects in the synthetic system

We display the ROC curve of our algorithm to identify recommended products in our synthetic simulation by varying the sparsity, $\gamma$ in $R_{\text{true}}$ (Figure 3a), varying $\alpha$ (Figure 3b), and varying exponent $e$ (Figure 3c) for acceptance probability. The dimensions of the rating matrix is fixed at [1000, 100] with 1000 users and 100 items. Decreasing $\alpha$ as well as $\gamma$ has adversarial effects on the ROC curve, and hence, AUC values, as is natural. The fact that high values of $\alpha$ produce more discriminative deconvolved ratings is clearly illustrated in Figure 2 f. Additionally, Figure 3 d shows that the calculated score varies linearly with the true score as we change the recommender exponent, $e$, color coded in the legend. Overall, our algorithm is remarkably successful in extracting recommended items from $R_{\text{obs}}$ without any additional information. Also, we can score the overall impact of the RS (see the upcoming section RS scores) and it accurately tracks the true effect of the RS.

## 3.4 Real data

In this subsection we validate our approach for deconvolving feedback loops on a real-world RS. First, we demonstrate that the deconvolved ratings are able to distinguish datasets that use a RS against those that do not. Second, we specify a metric that reflects the extent of RS effects on the final ratings matrix. Finally, we validate that the score returned by our algorithm is indicative of the recommender effects on a per item basis. We use $\alpha = 1$ in all experiments because it models the case when the recommender effects are strong and thus produces the highest discriminative effect between the observed and true ratings (see Figure 2 f). This is likely to be the most useful as our model is only an approximation.

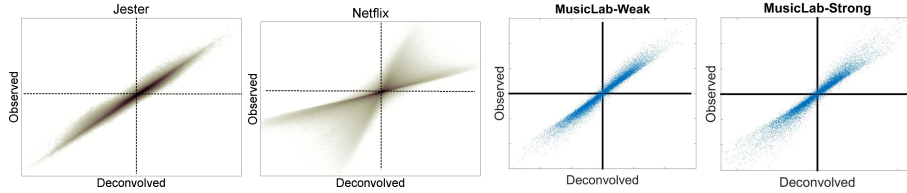

Figure 4: (Left to Right) A density plot of deconvolved and observed ratings on the Jester joke dataset (Left) that had no feedback loops and on the Netflix dataset (Left Center) where their Cinematch algorithm was running. The Netflix data shows dispersive effects indicative of a RS whereas the Jester data is highly correlated indicating no feedback system. A scatter plot of deconvolved and observed ratings on the MusicLab dataset- Weak (Right Center) that had no downloads counts and on the MusicLab dataset- Strong (Right) which displayed the download counts. The MusicLab-Strong scatter plot shows higher dispersive effects indicative of feedback effects.

**Datasets.** Table 1 lists all the datasets we use to validate our approach for deconvolving a RS (from [21, 4, 13]). The columns detail name of the dataset, number of users, the number of items, the lower threshold for number of ratings per item (RPI) considered in the input ratings matrix and the number of singular vectors $k$ (as many as possible based on the limits of computer memory), respectively. The datasets are briefly discussed in the supplementary material.

**Classification of ratings matrix.**

An example of the types of insights our method enables is shown in Figure 4. This figure shows four density plots of the estimated *true* ratings (y-axis) compared with the *observed* ratings (x-axis) for two datasets, Jester and Netflix. Higher density is indicated by darker shades in the scatter plot of observed and deconvolved ratings. If there is no RS, then these should be highly correlated. If there is a system with feedback loops, we should see a dispersive plot. In the first plot (Jester) we see the results for a real-world system without any RS or feedback loops; the second plot (Netflix) shows the results on the Netflix ratings matrix, which did have a RS impacting the data. A similar phenomenon is observed in the third and fourth plots corresponding to the MusicLab dataset in Figure 4. We display the density plot of observed (y-axis) vs. deconvolved or expected true (x-axis) ratings for all datasets considered in our evaluation in the supplementary material.

**Recommender system scores.** The RS scores we displayed in Table 1 are based on the fraction of ratings with non-zero score (using the score metric (10)). Recall that a zero score indicates that the data point lies outside the associated hyperbola and does not suffer from recommender effect. Hence, the RS score is indicative of the fraction of ratings affected by the recommender. Looking at Table 1, we see that the two Jester datasets have low RS scores validating that the Jester dataset did not run a RS. The MusicLab datasets show a weak effect because they do not include any type of item-item recommender. Nevertheless, the strong social influence condition scored higher for a RS because the simple download count feedback will elicit comparable effects. These cases give us confidence in our scores because we have a clear understanding of feedback processes in the true data. Interestingly, the RS score progressively increases for the three versions of the MovieLens datasets: MovieLens-100K, MovieLens-1M and MovieLens-10M. This is expected as the RS effects would have progressively accrued over time in these datasets. Note that Netflix is also lower than Movielens, indicating that Net-

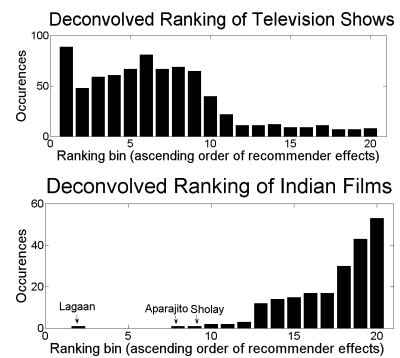

Figure 5: (Top to bottom) (a) Deconvolved ranking as a bar chart for T.V. shows. (b) Deconvolved ranking as a bar chart for Indian movies.

flix's recommender likely correlated better with users' true tastes. The RS scores associated with alcohol datasets (RateBeer, BeerAdvocate and Wine Ratings) are higher compared to the Fine Foods dataset. This is surprising. We conjecture that this effect is due to common features that correlate with evaluations of alcohol such as the age of wine or percentage of alcohol in beer.

**Ranking of items based on recommendation score.** We associate a RS rating to each item as our mean score of an item over all users. All items are ranked in ascending order of RS score and we

first look at items with low RS scores. The Netflix dataset comprises of movies as well as television shows. We expect that television shows are less likely to be affected by a RS because each season of a T.V. show requires longer time commitment, and they have their own following. To validate this expectation, we first identify all T.V. shows in the ranked list and compute the number of occurrences of a T.V. show in equally spaced bins of size 840. Figure 5 shows a bar chart for the number of occurrences and we see that there are $\approx$ 90 T.V.shows in the first bin (or top 840 items as per the score). This is highest compared to all bins and the number of occurrences progressively decrease as we move further down the list, validating our expectation. Also unsurprisingly, the seasons of the popular sitcom Friends comprised of 10 out of the top 20 T.V. seasons with lowest RS scores. It is also expected that the Season 1 of a T.V. show is more likely to be recommended relative to subsequent seasons. We identified the top 40 T.V shows with multiple (at least 2) seasons, and observed that 31 of these have a higher RS score for Season 1 relative to Season 2. The 9 T.V. shows where the converse is true are mostly comedies like Coupling, That 70's Show etc., for which the seasons can be viewed independently of each other. Next, we looked at items with high RS score. At the time the dataset was released, Netflix operated exclusively in the U.S., and one plausible use is that immigrants might use Netflix's RS to watch movies from their native country. We specifically looked at Indian films in the ranked list to validate this expectation. Figure 5b shows a bar chart similar to the one plotted for T.V. shows and we observe an increasing trend along the ranked list for the number of occurrences of Indian films. The movie with lowest recommendation score is Lagaan, the only Indian movie to be nominated for the Oscars in last 25 years.

## 4 Discussion, related work and future work

**Discussion:**In this paper we propose a mechanism to deconvolve feedback effects on RS, similar in spirit to the network deconvolution method to distinguish direct dependencies in biological networks [10, 3]. Indeed, our approach can be viewed as a generalization of their methods for general rectangular matrices. We do so by only considering a ratings matrix at a given instant of time. Our approach depends on a few reasonable assumptions that enable us to create a tractable model of a RS. When we evaluate the resulting methods on synthetic and real-world datasets, we find that we are able to assess the degree of influence that a RS has had on those ratings. This analysis is also easy to compute and just involves a singular value decomposition of the ratings matrix.

**Related Work:** User feedback in collaborative filtering systems is categorized as either explicit feedback which includes input by users regarding their interest in products [1], or implicit feedback such as purchase and browsing history, search patterns, etc. [14]. Both types of feedback affect the item-item or user-user similarities used in the collaborative filtering algorithm for predicting future recommendations [16]. There has been a considerable amount of work on incorporating the information from these types of user feedback mechanisms in collaborative filtering algorithms in order to improve and personalize recommendations [15, 6]. Here, we do not focus on improving collaborative filtering algorithms for recommender systems by studying user feedback, but instead, our thrust is to recover each user's true preference of an item devoid of any rating bias introduced by the recommender system due to feedback. Another line of work based on user feedback in recommender systems is related to understanding the exploration and exploitation tradeoff [20] associated with the training feedback loop in collaborative filtering algorithms [9]. This line of research evaluates *'what-if'* scenarios such as evaluating the performance of alternative collaborative filtering models or, adapting the algorithm based on user-click feedbacks to maximize reward, using approaches like the multi-armed bandit setting [17, 18] or counterfactual learning systems [5]. In contrast, we tackle the problem of recovering the true ratings matrix if feedback loops were absent.

**Future Work:** In the future we wish to analyze the effect of feeding the derived deconvolved ratings without putative feedback effects back into the RS. Some derivatives of our method include setting the parameters considered unknown in our current approach with known values (such as $S$) if known *a priori*. Incorporating temporal information at different snapshots of time while deconvolving the feedback loops is also an interesting line of future work. From another viewpoint, our approach can serve as a supplement to the active learning community to unbias the data and reveal additional insights regarding feedback loops considered in this paper. Overall, we believe that deconvolving feedback loops opens new gateways for understanding ratings and recommendations.

**Acknowledgements:** David Gleich would like to acknowledge the support of the NSF via awards CAREER CCF-1149756, IIS-1422918, IIS-1546488, and the Center for Science of Information STC, CCF-093937, as well as the support of DARPA SIMPLEX.

## Footnotes

[1]For an user-user similarities, $\hat{\boldsymbol{S}}$, the derivations in this paper can be extended by considering the expression: $\boldsymbol{R}_{\text{obs}}^{T} = \boldsymbol{R}_{\text{true}}^{T} + \boldsymbol{H}^{T} \odot (\boldsymbol{R}_{\text{obs}}^{T}\hat{\boldsymbol{S}})$. We restrict to item-item similarity which is more popular in practice.

[2]See [10] for details on scaling similarity matrices to ensure convergence

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
