[Supplementary Material]

# Supplementary Material for Deconvolving Feedback Loops in Recommender Systems

**Ayan Sinha**
Purdue University
sinhayan@mit.edu

**David F. Gleich**
Purdue University
dgleich@purdue.edu

**Karthik Ramani**
Purdue University
ramani@purdue.edu

This document serves as supplementary material to the paper: Deconvolving Feedback Loops in Recommender Systems.

## 1   Theorem and proof on deconvolving feedback loops

In this section, we prove equation 6 in the manuscript and provide a proof of the main theorem. Equation 6 in the paper is:

$$\hat{\boldsymbol{R}}_{\text{obs}} = \hat{\boldsymbol{R}}_{\text{true}}(\boldsymbol{I} + f_1(\alpha)\hat{\boldsymbol{R}}_{\text{true}}^T\hat{\boldsymbol{R}}_{\text{true}} + f_2(\alpha)(\hat{\boldsymbol{R}}_{\text{true}}^T\hat{\boldsymbol{R}}_{\text{true}})^2 + f_3(\alpha)(\hat{\boldsymbol{R}}_{\text{true}}^T\hat{\boldsymbol{R}}_{\text{true}})^3 + \cdots) \tag{1}$$

**Proof**   We use induction to first prove that:

$$\hat{\boldsymbol{R}}_{\text{obs}}^{k+1} = \hat{\boldsymbol{R}}_{\text{true}}(\boldsymbol{I} + \alpha_k\boldsymbol{S}_{\text{true}}(\boldsymbol{I} + \alpha_{k-1}\boldsymbol{S}_{\text{true}}\cdots(\boldsymbol{I} + \alpha_2\boldsymbol{S}_{\text{true}}(\boldsymbol{I} + \alpha_1\boldsymbol{S}_{\text{true}})^3)^3\cdots)^3) \tag{2}$$

Using assumptions 1,2, and 3, in the main manuscript we have:

$$\begin{aligned}
\hat{\boldsymbol{R}}_{\text{obs}}^2 &= \hat{\boldsymbol{R}}_{\text{true}} + \alpha_1(\hat{\boldsymbol{R}}_{\text{obs}}^1\boldsymbol{S}_1)\\
&= \hat{\boldsymbol{R}}_{\text{true}} + \alpha_1(\hat{\boldsymbol{R}}_{\text{true}}\boldsymbol{S}_{\text{true}})\\
&= \hat{\boldsymbol{R}}_{\text{true}}(\boldsymbol{I} + \alpha_1\boldsymbol{S}_{\text{true}}).
\end{aligned} \tag{3}$$

Here, $\hat{\boldsymbol{R}}_{\text{obs}}^2$ is the centered and normalized observed matrix after the first iteration, and $\hat{\boldsymbol{R}}_{\text{true}}$ is the matrix of true preferences which we wish to recover. We have used the initial conditions $\boldsymbol{R}_{\text{obs}}^1 = \boldsymbol{R}_{\text{true}}$, $\boldsymbol{S}_1 = \boldsymbol{S}_{\text{true}}$. For the second iteration, we have:

$$\begin{aligned}
\hat{\boldsymbol{R}}_{\text{obs}}^3 &= \hat{\boldsymbol{R}}_{\text{true}} + \alpha_2(\hat{\boldsymbol{R}}_{\text{obs}}^2\boldsymbol{S}_2)\\
&= \hat{\boldsymbol{R}}_{\text{true}} + \alpha_2(\hat{\boldsymbol{R}}_{\text{obs}}^2(\hat{\boldsymbol{R}}_{\text{obs}}^2)^T\hat{\boldsymbol{R}}_{\text{obs}}^2)\\
&= \hat{\boldsymbol{R}}_{\text{true}} + \alpha_2(\hat{\boldsymbol{R}}_{\text{true}}(\boldsymbol{I} + \alpha_1\boldsymbol{S}_{\text{true}})(\boldsymbol{I} + \alpha_1\boldsymbol{S}_{\text{true}})\hat{\boldsymbol{R}}_{\text{true}}^T\hat{\boldsymbol{R}}_{\text{true}}(\boldsymbol{I} + \alpha_1\boldsymbol{S}_{\text{true}}))\\
&= \hat{\boldsymbol{R}}_{\text{true}}(\boldsymbol{I} + \alpha_2((\boldsymbol{I} + \alpha_1\boldsymbol{S}_{\text{true}})^2\boldsymbol{S}_{\text{true}}(\boldsymbol{I} + \alpha_1\boldsymbol{S}_{\text{true}}))\\
&= \hat{\boldsymbol{R}}_{\text{true}}(\boldsymbol{I} + \alpha_2\boldsymbol{S}_{\text{true}}(\boldsymbol{I} + \alpha_1\boldsymbol{S}_{\text{true}})^3).
\end{aligned} \tag{4}$$

We have used the adjusted cosine similarity relationships, $\boldsymbol{S}_2 = (\hat{\boldsymbol{R}}_{\text{obs}}^2)^T\hat{\boldsymbol{R}}_{\text{obs}}^2, \boldsymbol{S}_{\text{true}} = (\hat{\boldsymbol{R}}_{\text{true}})^T\hat{\boldsymbol{R}}_{\text{true}}$, the property of matrix transpose $(\boldsymbol{AB})^T = \boldsymbol{B}^T\boldsymbol{A}^T$, and the property that $\boldsymbol{S}_{\text{true}}$ is symmetric, i.e., $\boldsymbol{S}_{\text{true}} = \boldsymbol{S}_{\text{true}}^T$.

The base cases are now proved in equations 3, 4. Considering equation 2 to be true at the $k^{th}$ iteration, we wish to prove the following at the $(k + 1)^{th}$ iteration:

$$\hat{\boldsymbol{R}}_{\text{obs}}^{k+2} = \hat{\boldsymbol{R}}_{\text{true}}(\boldsymbol{I} + \alpha_{k+1}\boldsymbol{S}_{\text{true}}(\boldsymbol{I} + \alpha_k\boldsymbol{S}_{\text{true}}(\boldsymbol{I} + \alpha_{k-1}\boldsymbol{S}_{\text{true}}\cdots(\boldsymbol{I} + \alpha_2\boldsymbol{S}_{\text{true}}(\boldsymbol{I} + \alpha_1\boldsymbol{S}_{\text{true}})^3)^3\cdots)^3)^3) \tag{5}$$

Using assumptions 1, 2 and 3 we have the following relationship:

$$\hat{\boldsymbol{R}}_{\text{obs}}^{k+2} = \hat{\boldsymbol{R}}_{\text{true}} + \alpha_{k+1}(\hat{\boldsymbol{R}}_{\text{obs}}^{k+1}\boldsymbol{S}_{k+1}). \tag{6}$$

We also have the adjusted cosine similarity relationship, $S_{k+1} = (\hat{R}_{obs}^{k+1})^T \hat{R}_{obs}^{k+1}$. Substituting this relation in equation 6, we have:

$$\hat{R}_{obs}^{k+2} = \hat{R}_{true} + \alpha_{k+1}(\hat{R}_{obs}^{k+1}(\hat{R}_{obs}^{k+1})^T \hat{R}_{obs}^{k+1}). \tag{7}$$

We substitute equation 2 in equation 7, and represent the matrix $(I + \alpha_k S_{true}(I + \alpha_{k-1} S_{true} \cdots (I + \alpha_2 S_{true}(I + \alpha_1 S_{true})^3)^3 \cdots)^3)$ as $M$. Matrix $M$ is symmetric, so we have $M = M^T$. As, $\hat{R}_{true}^T \hat{R}_{true} = S_{true}$, we get:

$$\begin{aligned} \hat{R}_{obs}^{k+2} &= \hat{R}_{true} + \alpha_{k+1}(\hat{R}_{true}M^2 \hat{R}_{true}^T \hat{R}_{true}M) \\ &= \hat{R}_{true} + \alpha_{k+1}(\hat{R}_{true}M^2 S_{true}M) \end{aligned} \tag{8}$$

Furthermore, matrices $M$ and $S_{true}$ have the same set of eigenvectors, and hence, we have:

$$\begin{aligned} \hat{R}_{obs}^{k+2} &= \hat{R}_{true} + \alpha_{k+1}(\hat{R}_{true}S_{true})M^3 \\ &= \hat{R}_{true}(I + \alpha_{k+1} S_{true}M^3) \end{aligned} \tag{9}$$

Substituting back for $M$, we get equation 5, completing the proof of equation 2 using induction. It is easy to verify that the maximum exponent of $S_{true}$ in $\hat{R}_{obs}^{k+1}$ grows exponentially as $3^{k-1}$, and we have the full set of powers of $S_{true}$. So even for a small number of iterations, we have an large number of terms in equation 2. As a result, if we expand equation 2 for a large number of iterations $k \to \infty$, we get the infinite series:

$$\begin{aligned} \hat{R}_{obs} &= \hat{R}_{true}(I + f_1(a_1)S_{true} + f_2(a_2)(S_{true})^2 + f_3(a_3)(S_{true})^3 + \ldots) \\ &= \hat{R}_{true}(I + f_1(a_1)\hat{R}_{true}^T \hat{R}_{true} + f_2(a_2)(\hat{R}_{true}^T \hat{R}_{true})^2 + f_3(a_3)(\hat{R}_{true}^T \hat{R}_{true})^3 + \ldots) \end{aligned} \tag{10}$$

Here, we replaced $\hat{R}_{obs}^{k+1}$ with the observed ratings matrix, $\hat{R}_{obs}$ after a large number of iterations. Also, $f_1, f_2, f_3 \ldots$ are functions of the probability parameters $a_k = [\alpha_1, \alpha_2, \ldots \alpha_k, \ldots]$ of the form $f_z(a_z) = c\alpha_1^{c_1}\alpha_1^{c_2} \cdots \alpha_k^{c_k} \cdots$, where $c$ is a constant. Also, as all $\alpha_k$'s appear as a multiplicative factor to $S_{true}$, we have $\sum_k c_k = z$ in $f_z(a_z)$. This completes our proof. ∎

The governing expression of our simplified recommender as stated in the main manuscript is:

$$\hat{R}_{obs} = \hat{R}_{true}(I + \alpha \hat{R}_{true}^T \hat{R}_{true} + \alpha^2 (\hat{R}_{true}^T \hat{R}_{true})^2 + \alpha^3 (\hat{R}_{true}^T \hat{R}_{true})^3 + \cdots) \tag{11}$$

We discuss the mechanism of solving equation 11 using the set of five assumptions in the main manuscript as a theorem.

**Theorem 1** *Assuming the recommender system follows* (11) *and the singular value decomposition of the observed rating matrix is,* $\hat{R}_{obs} = U\Sigma_{obs}V^T$, *the deconvolved matrix* $R_{true}$ *of true ratings is given as* $U\Sigma_{true}V^T$, *where the* $\Sigma_{true}$ *is a diagonal matrix with elements:*

$$\sigma_i^{true} = \frac{-1}{2\alpha\sigma_i^{obs}} + \sqrt{\frac{1}{4\alpha^2(\sigma_i^{obs})^2} + \frac{1}{\alpha}} \tag{12}$$

*where $\alpha$ is between 0 and 1.*

**Proof** Both $U, V$ are orthogonal matrices of dimension $m \times m$, and $n \times n$, respectively and $\Sigma_{true}$ is a non-negative diagonal matrix of singular values. Then, the eigenvalue decomposition of $S$ is given as:

$$S = \hat{R}^T \hat{R} = (U\Sigma_{true}V^T)^T U\Sigma_{true}V^T = V\Sigma_{true}^2 V^T. \tag{13}$$

Using assumption 5 and applying the Taylor series summation in equation 11, we have:

$$\hat{R}_{obs} = \hat{R}_{true}(I - \alpha S)^{-1}. \tag{14}$$

Note that in order for the series to converge, it is required that $\alpha(\sigma^{true})_i^2 < 1$ for all $i$. Below, we show that this holds for the choice of $\sigma^{true}$ in the theorem. In spectral form, we then have:

$$(I - \alpha S)^{-1} = V(I - \alpha\Sigma_{true}^2)^{-1}V^T. \tag{15}$$

Hence,

$$\hat{R}_{obs} = U\Sigma_{true}V^T V(I - \alpha\Sigma_{true}^2)^{-1}V^T = U\Sigma_{true}(I - \alpha\Sigma_{true}^2)^{-1}V^T. \tag{16}$$

Let the singular value decomposition of the observed rating matrix $\hat{\boldsymbol{R}}_{\text{obs}}$ be:

$$\hat{\boldsymbol{R}}_{\text{obs}} = \boldsymbol{U}\boldsymbol{\Sigma}_{\text{obs}}\boldsymbol{V}^T. \tag{17}$$

Using the (16) and (17) we find:

$$\boldsymbol{\Sigma}_{\text{obs}} = \boldsymbol{\Sigma}_{\text{true}}(\boldsymbol{I} - \alpha\boldsymbol{\Sigma}_{\text{true}}^2)^{-1} \quad , \text{i.e.} \quad \sigma_i^{\text{obs}} = \frac{\sigma_i^{\text{true}}}{1 - \alpha(\sigma_i^{\text{true}})^2} \quad \forall i. \tag{18}$$

Solving the quadratic in $\sigma_i^{\text{true}}$ in terms of $\sigma_i^{\text{obs}}$, and considering only the positive value, we get:

$$\sigma_i^{\text{true}} = \frac{-1}{2\alpha\sigma_i^{\text{obs}}} + \sqrt{\frac{1}{4\alpha^2(\sigma_i^{\text{obs}})^2} + \frac{1}{\alpha}} \tag{19}$$

What remains to show is that this choice of $\sigma_i^{\text{true}}$ satisfies the convergence assumption in the main manuscript. This can be inferred through the following inequalities:

$$
\begin{aligned}
\sqrt{\frac{1}{4\alpha^2(\sigma_i^{obs})^2} + \frac{1}{\alpha}} &< \sqrt{\frac{1}{\alpha}} + \sqrt{\frac{1}{4\alpha^2(\sigma_i^{obs})^2}} \\
\sqrt{\frac{1}{4\alpha^2(\sigma_i^{obs})^2} + \frac{1}{\alpha}} &< \sqrt{\frac{1}{\alpha}} + \frac{1}{2\alpha\sigma_i^{obs}} \\
\frac{-1}{2\alpha\sigma_i^{obs}} + \sqrt{\frac{1}{4\alpha^2(\sigma_i^{obs})^2} + \frac{1}{\alpha}} &< \sqrt{\frac{1}{\alpha}} \\
\sigma_i^{true} &< \sqrt{\frac{1}{\alpha}} \\
\alpha(\sigma_i^{true})^2 &< 1
\end{aligned}
\tag{20}
$$

■

## 2 Algorithms and codes for deconvolving feedback loops

The overall algorithm to assign a score for the extent of recommendation in a recommender system is shown below. The matlab code for deconvolving feedback loops is given below. The inputs to the

---

**Algorithm 1** Extract Recommended Ratings

---

**Input:** $\boldsymbol{R}_{\text{obs}}$, where $\boldsymbol{R}_{\text{obs}}$ is observed ratings matrix
**Output:** $s(\boldsymbol{R}_{\text{recom}})$, Likelihood score that item was suggested by recommender
1: Compute $\hat{\boldsymbol{R}}_{\text{true}}$ given $\boldsymbol{R}_{\text{obs}}$ using Algorithm 1
2: Plot $\hat{\boldsymbol{R}}_{\text{true}}$ vs. $\hat{\boldsymbol{R}}_{\text{obs}}$ for the ratings in $\hat{\boldsymbol{R}}_{\text{obs}}$
3: Calculate $m, c$ for $\hat{\boldsymbol{R}}_{\text{obs}}(:, i) = m\hat{\boldsymbol{R}}_{\text{true}}(:, i) + c$ for each item using RANSAC
4: Translate $[\hat{\boldsymbol{R}}_{\text{true}}, \hat{\boldsymbol{R}}_{\text{obs}}]$ by $c$ and rotate by $\pi/2 - m$ to obtain $[\breve{\boldsymbol{R}}_{\text{true}}, \breve{\boldsymbol{R}}_{\text{obs}}]$ that is approx. parallel to the y-axis.
5: Scale such that $\max(|\breve{\boldsymbol{R}}_{\text{true}}|) = \max(|\breve{\boldsymbol{R}}_{\text{obs}}|)$
6: $s(\boldsymbol{R}_{\text{recom}}) \leftarrow \text{real}\left(\sqrt{\breve{\boldsymbol{R}}_{\text{true}}^2 - \breve{\boldsymbol{R}}_{\text{obs}}^2}\right)$

---

algorithm are the observed ratings matrix, $\boldsymbol{R}_{\text{obs}}$ and parameter $\alpha$. The parameter $\alpha$ was set to 1 for all experiments on real datasets.

```
function Rdeconv = deconv(Robs,alpha)
 % Robs: a mxn matrix of observed ratings
 % alpha: parameter controlling the deconvolving of feedback loops
 % Rdeconv: a mxn matrix of deconvolved ratings

NR = sum(spones(Robs),2); % Vector of number of ratings of users
Sum_Rat = sum(Robs,2);    % Vector of sum of ratings of every user
Avg_Rat = Sum_Rat./NR;    % Vector of average rating of each user
Rmean = (sparse(diag(Avg_Rat))*ones(size(Robs))); % Full matrix of average ratings
```

```matlab
Rmean = Rmean.*(Robs>0); % Sparse matrix of average ratings for rated items
Robs_c = Robs-Rmean; % User centered ratings matrix

item_norms = (sum(Robs_centered.^2)); % Item norms
Robs_cn = Robs_c*sparse(1:length(item_norms),1:length(item_norms),...
          1./sqrt(item_norms),length(item_norms),length(item_norms));
          % User centered & normalized ratings matrix

[U,Sobs,V] = svd(Robs_cn); %Singular value decomposition

[~,~,singvalobs] = find(Sobs); %singular values
singvaltrue = -1./(2*alpha*singvalobs) + sqrt((1./(4*alpha^2*singvalobs.^2))...
          +1/alpha); %transformed values

Strue = sparse(1:size(Robs,2),1:size(Robs,2),singvaltrue,size(Sobs,1),...
      size(Sobs,2)); %Deconvolved singular values
Rdeconv = U*Strue*V'; %Deconvolved matrix

end
```

The matlab implementation for RANSAC adapted from http://en.wikipedia.org/wiki/RANSAC is given below. We set *num* = 2, *iter* = 100, *threshDist* = 0.1 and *inlierRatio* = 0.3 in all our experiments. RANSAC works better than least squares method for line fitting because it implicitly excludes outliers for line fitting.

```matlab
function [bestParameter1,bestParameter2] = ransac(data,num,iter,threshDist,inlierRatio)
% data: a 2xn dataset with #n data points
% num: the minimum number of points. For line fitting problem, num=2
% iter: the number of iterations
% threshDist: the threshold of the distances between points and the fitting line
% inlierRatio: the threshold of the number of inliers

number = size(data,2); % Total number of points
bestInNum = 0; % Best fitting line with largest number of inliers
bestParameter1=0;bestParameter2=0; % parameters for best fitting line
for i=1:iter
%% Randomly select 2 points
    idx = randperm(number,num); sample = data(:,idx);
%% Compute the distances between all points with the fitting line
    kLine = sample(:,2)-sample(:,1);
    kLineNorm = kLine/norm(kLine);
    normVector = [-kLineNorm(2),kLineNorm(1)];
    distance = normVector*(data - repmat(sample(:,1),1,number));
%% Compute the inliers with distances smaller than the threshold
    inlierIdx = find(abs(distance)<=threshDist);
    inlierNum = length(inlierIdx);
%% Update the number of inliers and fitting model if better model is found
    if inlierNum>=round(inlierRatio*number) && inlierNum>bestInNum
        bestInNum = inlierNum;
        parameter1 = (sample(2,2)-sample(2,1))/(sample(1,2)-sample(1,1));
        parameter2 = sample(2,1)-parameter1*sample(1,1);
        bestParameter1 = parameter1; bestParameter2 = parameter2;
    end
end
```

## 3  Validating our assumptions

The synthetic experiment gives us a quick way to validate our assumption. We consider two instances of the synthetic recommender system. The first (case A) is as described in the main manuscript simulating a real recommender system. In the second (case B), we always derive the item-item similarity from the users' true rating matrix according to equation 7 in the main

Figure 1: We plot the one-standard deviation distribution of effects on the mean difference in ratings, user means, and item norms as we run a recommender system using the true ratings matrix to generate the similarity compared to the observed ratings matrix. This plot shows a minimal effect around 0.2 for item norms, and around 0.05 for rating difference in the worst case due to our approximations and provides a rough justification for assumptions 3-5 in a synthetic setting.

manuscript. Thus, this experiment quantifies the overall effects of assumptions 3 and 4. We compute this difference in ratings for a number of different realizations and plot the distribution within one standard deviation of the mean as one line in Figure 1. For assumption 3 specifically, we compare the user-means of the ratings matrix of case A to the true user means and the item-norms of the matrix in case A to the true item norms. Distributions over these differences are the other two lines. We plot the evolution of these differences for 50 steps of the synthetic recommender system. This figure shows that the item norms do differ between these two realizations, but their difference is small enough to make the approximation plausible. Note that the user effects are bounded above by 5, and the item norm effects scale with the number of users. There were 1000 users, so the difference in item norms is very small compared with the number of users.

## 4 Real data

We briefly discuss the datasets we used in our experiments below.

***Jester.*** We use two versions of the Jester-joke dataset, one collected between April 1999 - May 2003 and the other between November 2006 - May 2009. These two datasets contain ratings directly reported by the users without a recommender system interface.

***MusicLab.*** MusicLab experiment was conducted at the Department of Sociology at Columbia University between 2004 and 2007 [3]. The motivation of the experiment was to investigate the influence of recommendations on the success or failure of a song. In real-time, participants arriving at the experiment were randomly assigned to one of two experimental conditions (1) weak and (2) strong social influence, which differed only by the availability of information on the past behavior of others. Participants chose which songs to listen to based solely on the names of the bands and their songs in the weak social influence condition, whereas participants could also see how many times each song was downloaded by previous participants in the strong social influence condition

[3]. Thus, these social influence conditions may be thought of as parallel realizations of a system with a weak-recommender and no-recommender.

***MovieLens.*** MovieLens data sets were collected by the GroupLens Research Project at the University of Minnesota. We use three versions of the data: MovieLens-100K consisting of one hundred thousand ratings, MovieLens-1M consisting of one million ratings and MovieLens-10M consisting of 10 million ratings. We gathered from the MovieLens website that Movielens-100K dataset was released on 4/1998, MovieLens 1M was released on 2/2003 and MovieLens 10M was released on 1/2009. Movielens 10M contains almost all movies in the 100K and 1M dataset. This indicates that the data corresponds to different snap-shots of the same data obtained at different times of the data collection process.

***Beer, Wine, Food.*** The beer-rating websites BeerAdvocate and RateBeer allow users to rate beers using a five-aspect rating system. They also include reviews of pubs. We also consider the wine review website CellarTracker, and reviews from the Fine Foods. Note that the rating datasets have a different scale (e.g. beers on RateBeer are rated out of 20, wines on CellarTracker are rated out of 100, etc.).

***Netflix.*** Netflix provided a training data set of 100,480,507 ratings that 480,189 users gave to 17,770 movies for the Netflix competition [1]. This dataset had the Cinematch algorithm running on it.

**Classification of ratings matrix.**

We display the density plot of observed (y-axis) vs. deconvolved or expected true (x-axis) ratings for all datasets considered in our evaluation in Figure 2. The two datasets that have no recommender system running on them are Jester-1 and Jester-2 [2]. Figures 2 a,b display the density plot of observed vs. deconvolved ratings for the two datasets. The fact that this dataset did not use a recommender system is evident from the density plot, wherein we see that the observed and deconvolved ratings are linearly correlated. In contrast, the density plot of observed vs. deconvolved ratings for the other datasets (Figure 2 c-k) show varying levels of dispersion, and indicate that the observed and deconvolved ratings are not very correlated or that a recommender system is operating on these datasets. It is the ratings that fall in the zones above and below the straight line of linear correlation that suffer from recommender effects.

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

Figure 2: Density plot of deconvolved vs.observed ratings for (a) Jester-1 (b) Jester-2 (c) MovieLens-100K (d) MovieLens-1M (e) MovieLens-10M (f) RateBeer (g) BeerAdvocate (h) Wine Ratings (i) Fine Foods (j) Netflix, respectively