[Reviews · NeurIPS 2016]

Reviewer 1

Summary

The authors tackle an interesting problem: studying how recommender system actually influence the consumer. They assume that the observed rating matrix is made of user true feeling and RS influence. Introducing sequence in the modeling, they consider the observation matrix as a fix matrix (user true feeling) and successive influences associated to a basic RS relying on item similarities. Finally they try to identify items that are affected by the RS. Then, they propose a metric to quantify RS effects and compare the results on different datasets (corresponding to real world website with ou without RS). The authors carry out some experiments on synthetic datasets to set the hyper-parameters of the method. This work is original and tackle a critical issue: measuring the efficiency of RS (persuasive force). Even if the measure explain the persuasive force to give better/lower ratings and not the real feelings of the users, this work is very interesting.

Qualitative Assessment

Regarding the scores, there seems to be a correlation with the item rating density (#rating/#items) (for films, alcohol and food)... Maybe there is a small bias in the procedure. The result on TV shows vs Indian films is amazing. Do you perform further analysis on the average rating in each case and on the consensus in the rating? The idea would be: if every one like an item, then no effect is associated to the RS. == typo l55 ans => and author should use a decent font for the references

Confidence in this Review

1-Less confident (might not have understood significant parts)


Reviewer 2

Summary

This paper systematically investigates the mechanism how much rating values are influenced by a RS (recommender system). The primary goal of this paper is to develop a method to infer the true rating matrix (which is not influenced by a RS) from the observed rating matrix, given by some plausible assumptions. The authors propose a truncated-SVD algorithm to recover the true rating matrix, and the score to measure how much the rating matrix is influenced by a RS. Even though the authors extensively validate the proposed algorithm on synthetic and real-world datasets, I believe that the assumptions raised by the authors does not properly model the real-world RS. The detailed questions are described in the below.

Qualitative Assessment

First of all, the problem considered in this paper is interesting and useful to some potential applications that require the true rating matrix not influenced by any recommender systems. However, the inference of the true rating matrix from the observed one is an ill-posed problem, which need the “strong” (somewhat unrealistic) assumptions. The questions about the assumptions are summarized as follows: 1. Assumption 1 is quite restricted in the sense that the popular recommendation algorithms (e.x. Bayesian matrix factorization) cannot be properly expressed in Eq. (2). If the real-world RS makes use of a complex recommendation algorithm that is not covered by the Assumption 1, it is hard to validate the quality of the true rating matrix extracted by the proposed algorithm. 2. In Assumption 2, the single Bernoulli random variable is used to approximate the indicator matrix (described in Assumption 1), which means that every user should follow the recommended rating by the same manner. I think that this is also non-realistic situation. Thus, I believe that Assumption 1 and 2 are too “strong” to model the real-world RS. Besides the assumptions, I have a question about the score described in Eq. (10), which is designed to measure how much the rating is affected by RS. I think that the proposed score is somewhat complex without proper reasons. For example, we simply use the distance between the observed ratings and the straight line as a score to measure the influence of RS. If the score should be normalized, a sigmoid function may be used to properly normalize the distance.

Confidence in this Review

2-Confident (read it all; understood it all reasonably well)


Reviewer 3

Summary

This paper mainly focuses on understanding whether it is possible to decompose the given observed rating matrix into user's true preference of items and the contribution by the recommendation system (RS). With strong model assumption, this paper proposes a singular value decomposition metric to recover user's true preferences from any instant rating matrix. Moreover, it also proposes a way to evaluate the influence of the RS on each user-item rating pair.

Qualitative Assessment

Generally I think this is a solid work to tackle a challenging and novel task. Understanding and quantify the feedback loop has not been studied by too many, and this work is a pioneer in this direction. The assumptions made in this work are not perfectly reasonable, but I can understand they are to some extend necessary to derive a theoretical solution. Same happens to the experiments. It is really hard to evaluate this work since the ground truth RS are hard to obtain. I feel the authors have tried their best to indirectly evaluate the model. The only concern lies in its practical impact. It is not clear what is the main usage of RS. For instance, it would have been more convincing if the authors can show that RS can be used to improve the rating prediction task.

Confidence in this Review

2-Confident (read it all; understood it all reasonably well)


Reviewer 4

Summary

The paper presents a methodology and evaluation for finding out which ratings in a dataset are likely to have been generated due to the influence of a recommender system. A simple model is proposed, under which users deviate from their "true" ratings over a series of steps. In each step (a) item similarity is used to make recommendations, and (b) users pick a recommendation and "adopt" it, adding it to their set of ratings, with a certain probability. The observed rating matrix is the limit of this process. The authors show that, under certain assumptions (e.g., knowledge of the adoption probability), the "true" rating matrix can be recovered from the "observed" ratings, i.e., its inference is tractable. The difference of the two is used by the authors as indication that certain items were recommended, rather than inherent to users. This intuition is applied and evaluated on a synthetic dataset, illustrating that this methodology can succesfuly discern "true" from "recommended" ratings. The same methodology is turned next to real datasets, where its validity is reinforced through anecdotal evidence: e.g., no recommended ratings are discovered in a dataset collected in the absence of a recommender system, while in a dataset including both movies and TV shows, tv shows were less likely to be recommended.

Qualitative Assessment

I appreciated the model and methodology proposed by the authors, as well as the difficulty of the task at hand. The authors do a very good job motivating their assumptions, and I have no objection with the idea that certain somewhat strong assumptions are needed to perform this. That said, I would have like to see a discussion on how this information (discerning "true" from "recommended" ratings) could actually be put to use. Why is this an important problem to solve? User preferences may be affected by a variety of exogenous factors; does accounting for this one lead to better recommendations? If this is the case, is a recommender, who has full view of the system, constrained by the same absence of information as assumed here? A technical question: the relationship between true and observed ratings ends up being determined by the singular value relationship (8) (or the cleaner (18) in the supplement). In short, it seems to be "boosting" items with high sigma^true_i. Is there some intuition to be gained from this? How is the algorithm identifying recommended items affected by the gap between sigma^true and sigma^obs, and can perhaps the inference occur on the singular value space, rather than by observing how ratings change? Section 3.2. is a bit abrupt: when stating "We develop an algorithm" it is not clear that his is an algorithm to classify items rated as "true" or "recommended". "validating that the Jester..." mention the fact that Jester did not use an RS earlier

Confidence in this Review

2-Confident (read it all; understood it all reasonably well)


Reviewer 5

Summary

In this paper, the authors aim to deconvolve the feedback loops in a recommender system. The authors propose an observation model with many assumptions, analyze it, and show that the deconvolution is possible.

Qualitative Assessment

- The practical usage/goal of this paper is not clear to me. Is the goal to understand to the effectiveness of a given recommender system? I suggest the authors to clarify the problem they try to solve and the practical impact of the problem. - Is there a way to improve recommendation performance if R_true is obtained? - Is the fact that most ratings are missing in R_obs modeled in the paper? Note that R_{true} should be a dense matrix with ratings (every user has a rating for each item, but it might not be revealed)

Confidence in this Review

2-Confident (read it all; understood it all reasonably well)


Reviewer 6

Summary

The goal of this work is to recover true ratings of intrinsic user preferences from the observed ratings. So sufficient assumptions are stated to deconvolve the feedback loops. This work furthermore develop a metric to unravel the influence of RS on the entire user-item rating matrix.

Qualitative Assessment

1. This work focuses only on the user-item rating matrix. So the item-item ( or user-user ) similiarities is the key to recover the true ratings. However, many real datasets include the temporal information, side information, contextual information, etc. It is more intresting and useful problem on how to recover the true ratings on these datasets. 2. There are too many materials stated in the supplementary file which destories the completeness of this work to some extent. 3. In line 55, "ans" should be "and"? 4. In line 148, "standard normal" should be "normal"?

Confidence in this Review

2-Confident (read it all; understood it all reasonably well)